# Epilepsy Due to Solitary Calcified Cysticercus Granuloma

**DOI:** 10.3390/pathogens12081037

**Published:** 2023-08-14

**Authors:** Jagarlapudi M. K. Murthy

**Affiliations:** Department of Neurology, CARE Hospitals, Banjara Hills, Hyderabad 500034, Telengana, India; jmkmurthy49@gmail.com

**Keywords:** neurocysticercosis, solitary calcified neurocysticercosis, solitary calcified cysticercus granulaoma, anti-seizure medication, focal-onset seizure, convulsive status epilepticus

## Abstract

The calcified stage of the neurocysticercosis (NCC) is the common cause of acquired epilepsy in low and middle income countries in people aged > 20 years. Approximately 30% of adult onset seizures and epilepsy are attributable to NCC. In India and some of the Latin American countries, epilepsy due to solitary calcified NCC is the common adult onset epilepsy. The current evidence suggests that the calcified cysticercus granuloma is probably the epileptogenic focus. The mechanisms involved in the epileptogenic process are not well understood; Focal-onset seizures with or without impaired awareness are the common seizure type. Focal-onset seizure can evolve to bilateral tonic-clonic seizure. Seizure outcome with anti-seizure medication, most often with monotherapy, is very good. The seizure disorders associated with various stages of NCC can be preventable.

## 1. Introduction

Neurocysticercosis (NCC), the infection of the human brain by the larvae of *Taenia solium*, is the leading cause of seizure(s) or epilepsy in low and middle income countries [1]. Pathological and observational cohort studies suggest that the calcified lesions seen on computed tomography (CT) scan, measuring less than 20 mm, represent the end result of the host’s inflammatory response to the larval form of *Taenia solium* [Figure 1A] [2,3,4]. The evolution of cysts phases lasts around 6 months on average, from the viable to calcified stage [5]. Several studies in the endemic countries for NCC suggest possible association between NCC and epilepsy. Two systematic reviews have found statistically significant association between NCC and epilepsy [6,7]. CT-based community studies in Atahualpa, Ecuador found that people with NCC had a high risk of having epilepsy and developing it [8,9]. Epilepsy due to solitary calcific NCC is a rare form of NCC and is mainly reported in India and in some South American countries [10,11]. In the countries endemic to NCC, calcified NCC is the commonest acquired cause of epilepsy in people aged > 20 years [7,10] and approximately 30% of adult-onset seizures and epilepsy are attributable to NCC [7,9]. This short review provides a comprehensive account of the epidemiology, clinical characteristics, and management of epilepsy due to solitary calcified neurocysticercosis and also possible mechanisms involved in the epileptogenesis.

## 2. Solitary Cysticercus Granuloma

A brief account about “Solitary Cysticercus (SCG)” is appropriate to ascertain a comprehensive overview of the subject. SCG resolves either spontaneously or with antihelminthic drugs or evolve into solitary calcific granuloma, which is the topic for discussion in this chapter. The concept SCG was first introduced in India in 1991 to describe a single, degenerating, contrast-enhancing parenchymal cysticercus cyst [3]. These are the most frequent neuroimaging abnormalities found in patients with new-onset seizures. This type of NCC is the most common type of NCC observed in India, and rarely reported from South American countries [4,10,11]. SCG represents the granular-nodular stage of cysticercus cyst. Diagnostic criteria were further defined subsequent period: clinical feature: focal-onset seizures with or without impaired awareness with or without bilateral tonic-clonic seizures; CT features a contrast-enhancing single small (<20 mm) and well-defined lesion; and MRI features are compatible with a diagnosis of SCG on a contrast CT scan [4]. In India, this pattern of NCC accounts for 50% to 80% of all forms of NCC [12,13,14], whereas in the South American countries endemic to NCC, this type of NCC accounts for about 20% of all types of NCC [15]. Patients with SCG in India are usually young teenagers or young adults presenting with new-onset seizures when compared to patients in South American countries [15]. Once larvae of *Taenia solium* lodges in the brain parenchyma, it typically undergoes through the four stages of involution: (a) vesicular, (b) colloidal, (c) granular, and (d) calcific stage. SCG is the granular–nodular form of the parenchymal cyst [14]. Under the new classification of seizures by the International League Against Epilepsy, seizures caused by this phase are categorized under “acute symptomatic seizures” and seizures caused by calcified NCC are categorized under unprovoked seizures [16]. A live cyst is generally asymptomatic, evoking minimal or no immune response. With the disintegration of the cyst, the parasitic antigen is exposed to the host’s immune system, and this results in an inflammatory response around the cyst. The subsequent formation of granulation tissue and edema around the parasite triggers seizures or headache in the host [17,18]. 

There are very few epidemiologic studies determining the prevalence and incidence of seizure disorders due to SCG. The quality of the studies was not of high standard [19,20]. It is extremely difficult to design high-quality epidemiologic studies of seizure disorders associated with NCC as the evolution of cysts’ phases lasts around 6 months on average, and the time period for each phase in an individual may vary [5]. Seizure(s) can be the presenting feature of all the evolutive stages of NCC. Contrast neuroimaging is mandatory for any NCC-related epidemiological studies. It is very difficult to estimate the optimal time for follow-up scans to detect the course of different evolutive stages seen in the initial CT scan. Short intervals between the onset of a clinical event and neuroimaging is probably associated with high-neuroimaging positivity. The prevalence of seizure disorders due to SCG was 0.14 (95%CI 0.08–0.26) per 1000 populations, probably a gross under estimation, as the time interval between the onset of seizures and neuroimaging was longer [19]. In a study in and around Hyderabad, the capital city of Telengana, a province in south India, the average annual incidence of seizure disorders due to SCG was 36.64 (95%CI 22.1–57.2) per 100,000. In this study, we conducted contrast CTs within a few weeks of onset of first seizure [20]. 

## 3. SCG—Natural Evolution—Seizure Relapse

During the natural evolution of SCG, the risk of seizure relapse is high in patients in whom the follow-up CT showed residual calcification. It was showed a high risk of seizure relapse in patients, in whom the first follow-up CT showed complete resolution of the lesions and the further follow-up CT showed calcification. In a prospective study, 215 patients with SCG and seizures, in whom anti-seizure medication (ASM) was tapered with the resolution of the lesion in the follow-up CT scan, were followed-up for relapse of seizures. All patients with recurrence of seizures and those with a minimum follow-up > 2 years after ASM withdrawal were included in the analysis. Of the 185 patients, 34 (18.3%) patients had calcification on the repeat CT brain scan; 10 (29.4%) of them had seizure relapse. Patients who had calcification had a 2.7 times (*p* = 0.01) higher risk of having recurring seizures, as compared to patients without calcification [21]. A systematic review of retrospective studies of patients with seizures, due to SCG, showed the presence of a calcific residue in the natural evolution of SCG, thus increasing the risk of seizures in the post-resolution period (pooled OR 11.27; 95% CI 4.96–26.51; *p* = 0.0001) [4].

## 4. SCG: Risk Factors for Calcification

Review of observational studies suggests that the natural history of SCG could take one of the following paths: (1) complete or (2) partial resolution of the lesion leaving a punctuate calcific speck(s). In individuals with SCG infection not exposed to the cycticidal drugs, the reported rates of spontaneous resolution of the lesions in the one-year follow-up contrast CTs were 63% [22] and 45% [23]. In the Vellore, South India study in individuals with SCG infection not exposed to cycticidal drugs, the rates of spontaneous resolution were 19% at 3 months, 36% at 6 months, and 63% at 1 year [22]. The precise mechanisms of calcification are undetermined. Of the 497 patients that participated in the three trials of antiparasitic treatment, the overall proportion of cyst calcification was 38% (188/497 cysts, from 147 patients). Predictors for calcification at the cyst level were cysts larger than 14 mm, and cysts with edema at baseline. At the patient level, having had more than 24 months with seizures, mild antibody response, increased dose albendazole regime, lower doses of dexamethasone, not receiving early antiparasitic retreatment, or complete cure were associated with an increased risk of calcification [24].

## 5. Calcified NCC: Epileptogenesis

In the endemic regions, calcific NCC is a common finding in individuals with seizure(s)/epilepsy than those without. Current evidence suggests a strong association between calcific NCC and epilepsy [8,9,10,25]. This association is also supported by the prevalence and incidence studies. A CT-based community study in rural Ecuador showed a three-fold increase in those having calcific NCC than those without seizures [8]. In the same community, the incidence study showed that individuals with NCC were six times more likely to develop adult-onset epilepsy than those without this disease after adjusting for relevant covariates [9]. The earlier calcific stage was considered as an ‘inactive disease’ [15,25]. However, the accumulating evidence suggests that these lesions substantially contribute to the development and maintenance of seizures [25]. Perilesional edema (PEO) close to the seizure is a common finding in patients with calcified NCC [26]. The edema and contrast enhancement of the calcific lesion may suggest that the calcific lesion might be an epileptogenic lesion [Figure 2 and Figure 3] [25,26,27,28]. 

Fujita and colleagues imaged a marker of neuroinflammation, translocator protein, using positron emission tomography (PET) and the selective ligand [^11^C]-PBR28 in nine patients with perilesional edema, degenerating cyst or both. [^11^C]-PBR28 is a PET radiotracer that binds to a 18pkD translocator protein (TSPO). The results of the study showed an increased TSPO in perilesional edema, which indicates an inflammatory etiology [29]. A prospective study in patients with new-onset seizures due to SCG using serial magnetization transfer imaging showed that patients with perilesional gliosis had a high-seizure burden and higher risk of seizure recurrence [30]. 

The histopathological descriptions of two calcified NCCs associated with perilesional edema episodes showed degenerating but recognizable cestode larval structures with a pronounced inflammation consisting of mononuclear lymphocytes, macrophages, and some eosinophils, as well as astrogliosis and microgliosis in the surrounding brain parenchyma [31,32]. In a high-volume epilepsy surgery center in south India, 17 patients with drug resistant epilepsy due to calcified NCC were identified. Eleven patients were advised to undergo respective surgery and five patients received this; in all of them, the lesion was solitary and extratemporal. In all of the patients, there was a concordance between the epileptogenic zone determined from presurgical noninvasive data and location of the lesion on MRI. Twelve (66.7%) of these lesions had perilesional gliosis on T2-weight and FLAIR MRI sequences. Of the five patients who had surgical excision, four patients were completely seizure free. Histopathologic examination of the lesions showed dense calcifications by chronic inflammatory infiltrates and reactive gliosis in surrounding brain surrounded parenchyma [Figure 1B and Figure 4] [33].

The evidence provided above suggests that the solitary calcified NCC is possibly the epileptogenic lesion. Ongoing chronic neuroinflammation incited by parasitic remnants or antigens within the calcification and associated blood–brain–barrier (BBB) impairment and perilesional gliosis are the two important players in the epileptogenesis. Neuroinflammation appears to affect seizure severity and recurrence. Recent studies showed that seizures also can increase the BBBs permeability, which can intensify and perpetuate neuroinflammation [34]. 

## 6. Epilepsy due to Solitary Calcified NCC: Epidemiology

In India, solitary calcified NCC is the common lesion in individuals with adult-onset epilepsy [10,35,36]. The reported prevalence rates of epilepsy due to this lesion in India ranged from 0.84 to 0.99 per 1000 population [10,37,38]. Additionally, the reported average annual incidence rate of epilepsy due to solitary calcified NCC in children was 9.64 (95% CI 3.1–22.5) per 100,000 populations. The study population included very highly selected school-going children seeking education in government primary schools in the suburbs around Hyderabad, the capital city of the Telengana state, a province in south India [39]. However, to establish the etiology of a chronic disease like NCC, it is mandatory to carry out a prospective longitudinal study of cohorts with incident cases to establish the temporal cause–effect relationship. For studies with prevalent cohorts, it is not possible to establish this relationship.

## 7. Epilepsy due to Solitary Calcified NCC: Clinical Characteristic

The common seizure type is focal-onset with or without impaired awareness. Other seizure types are focal-onset to bilateral tonic-clonic seizure, focal myoclonic seizure, and unknown-onset motor to bilateral tonic-clonic seizure. Solitary calcified NCC is the most common etiology of adult-onset epilepsy in India. In our recent community-based study, solitary calcified NCC accounted for 41% of the established etiology among 451 prevalent cases of epilepsy [10]. In our university hospital-based study, the mean age at presentation was 20 years [35], and it was 28.87 years in the rural community-based study performed 25 years apart [10]. In most of the studies conducted across India, there was no significant gender difference. Previous history of epileptic seizure(s) can be a fatal. In our university hospital-based study, 18 (18.5%) patients had a previous history of unprovoked seizures [35]. A patient may present with an isolated seizure, seizure cluster, initial high-seizure frequency, and convulsive status epilepticus (CSE) [10,35,36]. Convulsive SE is extremely rare in patients with epilepsy due to solitary calcified NCC. In a university hospital-based neurological intensive care unit study in an endemic area, of the 41 patients admitted for CSE due to different evaluative stages of NCC over 18 years, epilepsy, due to solitary calcified NCC, accounted for 20 (48.7%) of patients. The duration of CSE was shortest in patients with epilepsy, due to solitary calcified NCC, when compared to CSE duration in patients due to other evaluative stages. (1.96 h; range 0.5–5 h; *p* < 0.027) [40].

The clinical seizure pattern is clearly distinctive, as to allow it to be localized to the site of lesion on CT scan in 26% of patients [35]. Lesion location was more common in the peri-rolandic, followed by frontal and least in temporal [10,35,36]. Seizure semiology is frequently concordant with the localization of the calcified lesion in 58% to 74% of the patients [10,35]. Concordance between electro-clinical and radiological data in localizing and lateralizing was reported in 26–55% of patients, and the discordance in the remaining patients may be related to the epileptic network and seizure propagation patterns [35,36].

## 8. Anti-Seizure Medication 

Long-term seizure remission can be achieved in about 80% of patients often with monotherapy. Drug resistance is extremely rare and surgery is beneficial for patients with discrete lesion(s). All the four patients with seizure semiology indicated that the temporal-lobe origin had poor drug response. Most patients who had seizure recurrence following ASM withdrawal would continue to maintain good seizure control with reintroducing the appropriate ASMs [35]. In our recent rural community-based prevalent cohort, ≥2-year seizure remission was 80.3% and the independent predictor of drug resistance was due to the failure to respond to monotherapy (odds ratio: 63.9; 95% CI: 8.4–485.4; p b 0.0001) [10]. In our first university hospital-based study, 71.5% (95% CI: 7–85.4) of patients achieved 3-year remission, and 66% (95% CI: 32.4–88.2) achieved 5-year remission [36]. In a large volume epilepsy surgery center in south India, 35 (0.9%) of 3895 patients with drug-resistant epilepsy had epilepsy due to calcific stage [33].

## 9. Neurocysticercosis: Epilepsy Surgery

Drug resistant epilepsy, requiring epilepsy surgery in patients with NCC, is extremely uncommon. A systematic review identified three distinct subgroups: (1) cysticercotic epileptogenic lesion; (2) dual pathology including cysticercotic lesion and hippocampal sclerosis; and (3) cysticercotic lesion not related to the epileptogenic lesion [41]. Of the three surgical subgroups, surgical outcomes are strong in patients in the first two subgroups. There should be good concordance between seizure semiology, ictal EEG, and neuroimaging. 

## 10. Conclusions and Feature Research

The precise mechanisms involved in the epileptogenesis of the calcified cysticercus granuloma were not defined. Data available from the clinical, radiological, clinic-radiological, and scanty pathological studies suggest the possible role of chronic neuro-inflammation and perilesional gliosis, among the other factors [42]. Ooi et al. reported a patient with calcified cysticercus granuloma and recurrent seizures and PEO. Histopathology of PEO showed features of inflammation, suggesting an inflammatory etiology for PEO, which may be caused by intermittent release of antigen from the calcified parasite [32]. The study by Gupta et al. demonstrated scolex in all the 14 patients with PEO and rim enhancement on gradient echo imaging [43]. Perilesional gliosis was demonstrated in the operated specimens of patients with drug resistant epilesy [33]. Future research should include detailed pathological. genetic studies and the development of good experimental animal models.

## 11. Highlights

In endemic countries, calcified NCC is the most common cause of adult-onset epilepsy.

Solitary calcified NCC is the most common type of NCC in India compared to South American countriesFocal–onset with or without impaired consciousness and focal-onset to bilateral tonic-clonic seizures are the most common seizure typeThe calcified cysticercus cyst is probably the epileptogenic focus and the mechanisms involved in the pathogenesis are not well understood.Seizure remission rate is about 80%, more often with single anti-seizure medication in people with epilepsy due to solitary calcified NCC

## Figures and Tables

**Figure 1 pathogens-12-01037-f001:**
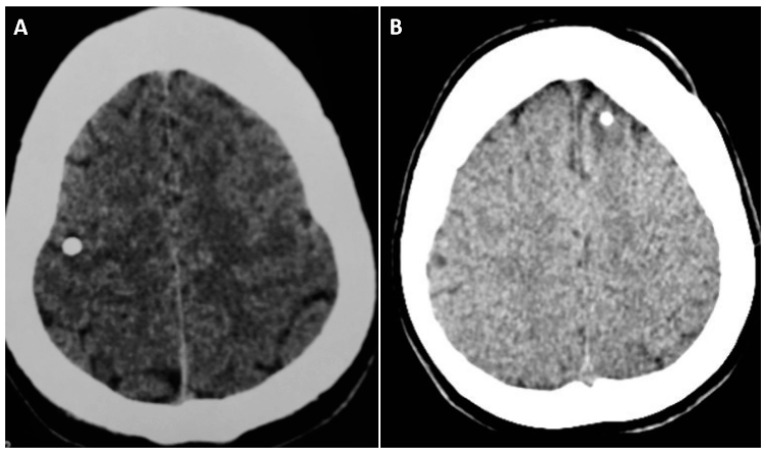
(**A**) Non-contrast CT scan showing a solitary hyperdense calcific lesion in the right frontal lobe in an adult woman with left focal-onset impaired awareness seizures; (**B**) Non-contrast CT scan showing a calcific lesion in the left frontal lobe with perilesional gliosis in a girl with anti-seizure medication resistant epilepsy.

**Figure 2 pathogens-12-01037-f002:**
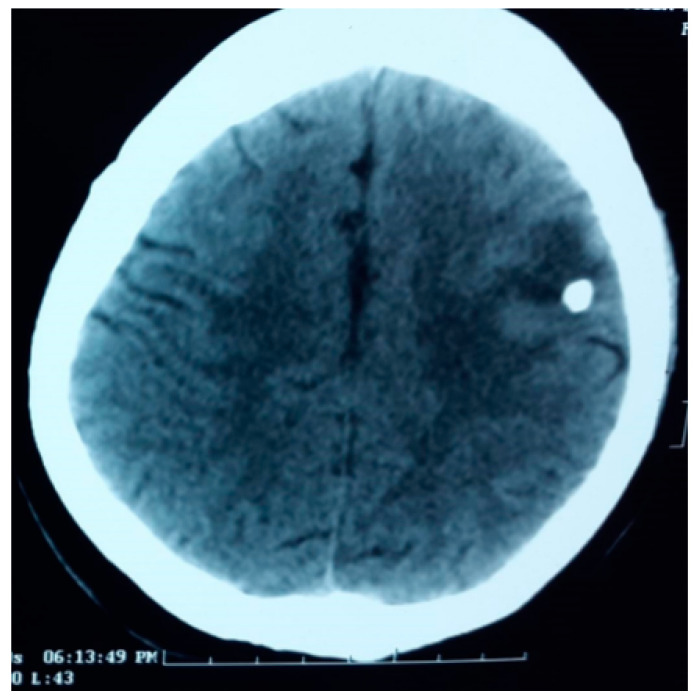
Non-contrast CT scan in an adult showing a round calcific lesion in the left frontal lobe with area of hypodensity anterior to the lesion, he had a recent history of seizures.

**Figure 3 pathogens-12-01037-f003:**
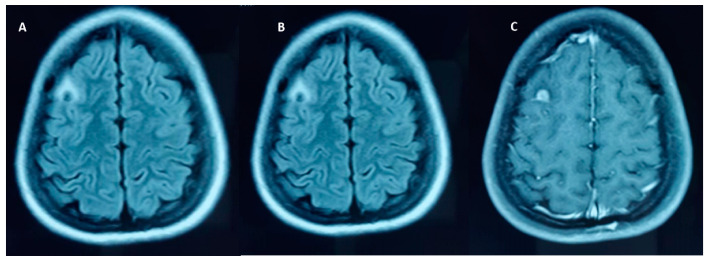
FLAIR Hypointense focus in the right frontal lobe surrounded by a rim of hyperintensity, which is in turn surrounded by perilesional edema. Post contrast study shows nearly solid appearing enhancement of the lesion (**A**–**C**).

**Figure 4 pathogens-12-01037-f004:**
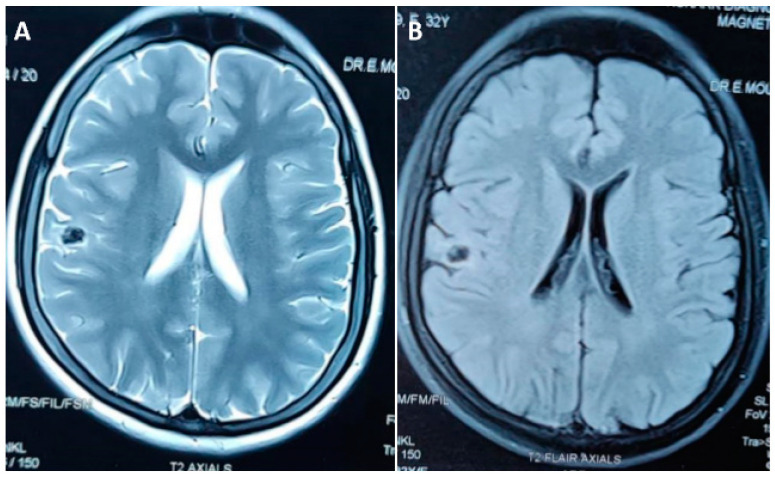
Non-contrast MRI scan showing a hypointense lesion on T2 (**A**) and fluid attenuated inversion recovery (**B**) images in the right frontal lobe laterally with the surrounding areas showing volume loss and T2 and FLAIR hyperintensity representing gliotic changes.

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
