# Peer review of "Epilepsy Due to Solitary Calcified Cysticercus Granuloma"

_pathogens, 2023, doi:10.3390/pathogens12081037_

Round 1

Reviewer 1 Report

This revision concerning neurocysticercosis is well conducted and very well documented, with references concerning India and other Asian countries, but also Latin America. Consequently, in my opinion, thanks to the interest of neurocysticercosis and the scientific content presented in this Ms, it deserves to be published.

However, I have certain concerns regarding some shortcomings when it comes to the format and the presentation of this review:

-          Reviews of pathogens must indentify current gaps and problems, and they should be critical and constructive and provide recommendations for future research; in the present case, the authors did not reach all of these aims;

-          The review itself lacks objectives which could help to introduce the reader in a better way to the content of the Ms; moreover, this review is not presented as a review;

-          Discussion as well Future Directions sections should be included to improve the review;

-          The Conclusions seem to be another abstract;

-          The reference list needs a revision of its format;

-          The references in the text will be written without a “.” before all of them; if a punctuation mark is needed, it will be placed after the reference, never before (the correct way thus is “features were defined later [4]. The author’s way is “In India this lesion features were defined later. [4] In India this lesion”);

-          Scientific names should be abbreviated, if they are not at the beginning of a sentence, after their second appearance, for example T. solium; they should always be in italics, even in the reference list;

-          The following abbreviations are not needed, they appear only once in the text, ILAE, CRESSI, TSPO; moreover, the definition of the abbreviation FLAIR is missing;

-          The English grammar needs to be revised.

Author Response

responses uploaded as separate file.

Reviewer 2 Report

This review doesn’t provide any additional information in comparison with those several similar reviews that has been published so far. Anyway, the bottom line of any analysis regarding the relationship between NCC and seizures/epilepsy should be based on the following reasoning or premises:

So far, there are not reliable incident studies (observational longitudinal cohorts) addressed to look at risk factors for epilepsy in patients with NCC (including well designed control groups, statistically significant samples, etc.)

Not all calcifications correspond to neurocysticercosis. There are several other etiologies showing single or multiple calcifications in neuroimaging, such as calcifications due to vascular, neoplastic, endocrine, viral, as well as other parasitic and CNS infections.

Surprisingly the authors do not take into consideration the above-mentioned reasoning and do not even mention it

Most of statements in the abstract section are personal opinions but they are not based on evidence

In the abstract section, as well as in the introduction and highlights sections, it is stated that “calcified NCC is the most common cause of adult-onset epilepsy” This is not true, most studies that suggest this statement are based on cross-sectional studies, addressed to look at prevalence of epilepsy, which are no the appropriate studies to analyze cause/effect between NC and epilepsy. These studies are biased because do not differentiate between acute symptomatic seizure and epilepsy (recurrence of seizures), as recommended by the ILAE.  To my knowledge, there are not prospective studies showing that calcifications/NCC is a risk factor for epilepsy (otherwise, the authors should cite the respective references). 

Throughout the manuscript, it is mentioned that “Current data from several types of studies: epidemiological, clinical, and radiological, suggest strong association between calcified NCC and epilepsy”. However the references cited by the Author to support this statement are reviews based on just opinions of their authors or merely anecdotal reports with evident methodological limitations

All studies listed by the author trying to substantiate the objective of this review are cross-sectional studies addressed to look at the prevalence of epilepsy, which are not appropriate to determine causal effect.  Additionally,  these studies are seriously biased because they have been done in highly selected population (for example the study carried out in Ecuador), therefore those studies cannot in any way be generalized to the entire population, not even to low-income countries.  Worse still middle or high-income countries

Just an example, on the “Calcified NCC: Epileptogenesis”, it is stated “A recent study imaged a marker of neuroinflammation, translocater protein (TSPO) using …. The interpretation of the PET findings is consistant with an inflammatory etiology of the per-ilesional edema” This report (reference 27)  was done ten years ago in NINE PATIENTS!!!! This study as well as few other similar ones published at that time have not been replicated to date.

The authors conclude that “NCC (or calcifications) is the leading cause of epilepsy worldwide. This is an inacceptable statement. In high-income and most of middle-income countries, NCC doesn't even exist. So, how can NCC be the main cause of epilepsy.? I invite the author to ask neurologists from a high-income country if they have ever seen a patient with NCC, the answer will surely be: never

Author Response

responses uploaded as separate file

Reviewer 3 Report

This review described several studies related to the casuistry of cysticercosis in India, which is particularly different from South America. 

The review is really interested and provides a summary of relevant information. 

Some minor comments are:

It is necessary to clarify some paragraphs that are not entirely clear.

In Solitary Cysticercus Granuloma:

Could you give more details about this idea? Why is extremely difficult and the estimates are not true?

"It is extremely difficult to design high quality epidemiologic studies with this lesion because of the natural evolution of the parasite. The projected estimates may not be true estimates."

The following sentence is not clear, require revision.

"The Comprehensive Rural Epilepsy Study in South India (CRESSI) in a rural community the prevalence of seizure disorder due to SCG was 0.14 (95%CI 0.08-0.26) per 1,000 populations, probably a gross under estimation"

In Calcified NCC: Epileptogenesis

This sentence seems not complete

"A recent study imaged a marker of neuroinflammation, translocater protein (TSPO) using positron emission tomography (PET) the selective ligand 11C-PBR28"

In Epilepsy due to Solitary Calcified NCC: Clinical Characteristics

The last paragraph is repeated:

"Focal-onset with or without impaired awareness is the most common seizure type. Lesion location was more common in the peri-rolandic followed by frontal and least in the temporal. [6,33,34] Seizure semiology is frequently concordant with the localization of the calcified lesion in 58% to 74% of the patients.[6,33] Concordance between electroclini-cal and radiological data in localizing and lateralizing was reported in 26% -55% of pa-tients and the discordance in the remaining patients may be related to the epileptic net-work and seizure propagation patterns. Forty minute awake and sleep EEG recording may be normal. [6,35]

The clinical seizure pattern can be clearly distinctive to allow it to be localised to the site of lesion on CT scan in 26% of patients.[33] Lesion location was more common in the peri-rolandic followed by frontal and least in temporal.[6,33,34] Seizure semiology is fre-quently concordant with the localization of the calcified lesion in 58% to 74% of the pa-tients.[6.34] Concordance between electroclinical and radiological data in localizing and lateralizing was reported in 26% -55% of patients and the discordance in the remaining patients may be related to the epileptic network and seizure propagation patterns..[34.35]"

In Anti-seizure Medication

Complete the idea:

"Drug resistance is extremely."

In the whole text:

- Please check the acronyms: Some are used only once and it is not necessary to use them like: BBB in the abstract, ILAE, CRESSI. BBB is defined twice in the manuscript and ASM is not defined.

- the citations seem to be misplaced throughout the text. The point appears first and then the citation numbering.

- Punctuation between sentences and use of capital letters should be checked in the manuscript.

Author Response

responses uploaded as separate file

Round 2

Reviewer 2 Report

The author insists on stating that cross-sectional studies of NC are evidence that this disease is the main cause of NC. This is a question of basic knowledge of epidemiology. To establish the etiology of a chronic disease, it is mandatory to carry out a prospective longitudinal study of cohorts with incident cases (not prevalent, since it is not possible to establish the temporal cause-effect relationship). Briefly, to demonstrate that NC calcifications are a potential risk factor for acquiring epilepsy, it would be necessary to select a population of cases with first seizures and CP calcifications, as well as select a population of patients with first seizures without calcifications, to make comparisons. These two population groups should be prospectively followed for several years to compare seizure recurrence (ie, epilepsy). According to the ILAE definition, there should be a risk of recurrence greater than 60% in the NC group to conclude that NC calcifications are a risk factor for epilepsy.

To the best of my knowledge, this study design has not been carried out thus far. Thus, the author should mention this gap in the knowledge of NC and recommend that these studies be done in the future to clarify once and for all if NC is really a risk factor for epilepsy (surely it must be, but it must be proven).

By the way, the author states on his reply  “No Journal will accept any research paper on any aspect of NCC without neuroimaging”.  I have never said otherwise, it seems that the author misunderstood my comment about it.

Author Response

Response

I understood your point of you. I have added information on this issue at the end of a paragraph on “epilepsy due to calcified NCC: Epidemiology” on page 9.

Thank you